# Patients’ and Healthcare Professionals’ Experiences and Views of Recurrent Urinary Tract Infections in Women: Qualitative Evidence Synthesis and Meta-Ethnography

**DOI:** 10.3390/antibiotics12030434

**Published:** 2023-02-22

**Authors:** Leigh N. Sanyaolu, Catherine V. Hayes, Donna M. Lecky, Haroon Ahmed, Rebecca Cannings-John, Alison Weightman, Adrian Edwards, Fiona Wood

**Affiliations:** 1Division of Population Medicine, Cardiff University, Cardiff CF14 4YS, UK; 2Primary Care and Interventions Unit, UK Health Security Agency (UKHSA), Gloucester GL1 1DQ, UK; 3School of Medicine, Cardiff University, Cardiff CF14 4XN, UK; 4Centre for Trials Research, Cardiff University, Cardiff CF14 4YS, UK; 5Specialist Unit for Review Evidence, Cardiff University, Cardiff CF14 4YS, UK

**Keywords:** qualitative evidence synthesis, meta-ethnography, recurrent urinary tract infection, qualitative research, patient experience, healthcare professionals’ experience

## Abstract

Background: Urinary tract infections (UTIs) are a common and significant problem for patients, clinicians, and healthcare services. Recurrent UTIs (rUTIs) are common, with a 3% prevalence in the UK. Although acute UTIs have a significant negative impact on the lives of patients, evidence of the impact of rUTIs is limited. To enhance shared decision-making around rUTI management, it is important to understand both the patients’ and healthcare professionals’ (HCPs’) perspectives. The objective of this qualitative evidence synthesis is to understand patients’ and HCPs’ experiences and views in the management of rUTIs. Methods: A qualitative evidence synthesis (QES) was performed that included primary qualitative studies involving patients with rUTIs or primary care HCPs who manage patients with rUTIs, up to June 2022. The following databases were searched: MEDLINE, Embase, CINAHL, PsycInfo, ASSIA, Web of Science, Cochrane Database of Systematic Reviews, Epistemonikos, Cochrane Central Registry of Controlled Trials, OpenGrey, and the Health Management Information Consortium (HMIC). The QES was prospectively registered on PROSPERO (CRD42022295662). Reciprocal translation was conducted and developed into a line of argument synthesis. We appraised the confidence in our review findings by using GRADE-CERQual. Results: Twelve studies were included in the final review; ten of those included patients, and three included HCPs (one study included both). Our review demonstrates that women with rUTIs have a unique experience, but it is generally of a chronic condition with significant impacts on numerous aspects of their lives. Antibiotics can be “transformative”, but patients have serious concerns about their use and feel non-antibiotic options need further research and discussion. HCPs share similar views about the impacts of rUTIs and concerns about antibiotic use and find the management of rUTIs to be complex and challenging. Based on our GRADE-CERQual assessment of the review findings, we have moderate confidence in those related to patients and low confidence in those related to HCPs. New conceptual models for both patients and HCPs are presented. Conclusions: This review has significant clinical implications. Patients require information on antibiotic alternative acute and preventative treatments for rUTIs, and this is not currently being addressed. There are communication gaps around the impact of rUTIs on patients, their perceived expectation for antibiotics, and the reasons for treatment failure. Further development of current clinical guidance and a patient decision aid would help address these issues.

## 1. Introduction 

Urinary tract infections (UTIs) are a common and significant problem for patients, clinicians, and healthcare services. Acute UTIs have a negative impact on patients’ daily lives, with reductions in activity, quality-of-life, and work attendance [1,2,3,4,5,6,7]. Recurrent UTIs (rUTIs), defined as two or more UTIs in six months, or three or more in 12 months, are common, with an estimated 3% annual prevalence in women in the UK [1,8,9,10]. Evidence of the impact of rUTIs is limited, and a recent review highlighted the need for more research on the experiences of women with rUTIs, given that they likely affect women in different ways [7].

The National Institute for Health and Care Excellence (NICE) has published guidelines on shared decision-making (SDM) to ensure that this is central to care in the NHS [11]. To enhance SDM around rUTI management, it is important to understand the perspectives of both patients and healthcare professionals (HCPs). SDM involves a joint discussion about treatment choices that includes all available options (including the option for no treatment) and exploring these in the context of the patient's personal preferences [11,12,13,14,15,16]. It is therefore important to understand “what matters most” to patients, their information needs, and concerns during this shared decision process [11,12,13,14,15,16]. Qualitative research methods are best suited to gain an in-depth understanding of these factors. 

Qualitative evidence synthesis (QES) is a systematic method that aims to bring together the findings from individual qualitative studies to provide an enhanced understanding about the phenomenon of interest [17]. Meta-ethnography is a method of QES that aims to generate new interpretations and conceptual understanding of the phenomenon of interest rather than amalgamate and describe the results of the included studies [17,18,19,20]. We aim to use these new interpretations to provide a greater understanding of the perspectives of patients with rUTIs and the HCPs managing their conditions. 

The objective of this QES is to understand patients’ and HCPs’ experiences and views in the management of rUTIs. 

## 2. Methods

We report a meta-ethnography with reference to the eMERGe guidelines and the seven phases described by Noblitt and Hare (Appendix A) [18,21]. We prospectively registered our QES on PROSPERO (CRD42022295662, available from https://www.crd.york.ac.uk/prospero/display_record.php?ID=CRD42022295662 accessed on 19 January 2022). The first phase of meta-ethnography reporting, “getting started”, is described above in terms of the rationale for the QES and its objectives [18,21]. Hereafter, the relevant phases are highlighted within each section. 

### 2.1. Search Strategy 

The search strategy was developed, piloted, and refined (LNS and AW) by using three key studies identified in preliminary searches [22,23,24]. Search terms related to the concepts of UTIs, female sex, and qualitative study methods were combined by using the Boolean operator “AND”. There were no restrictions on date or language. The search strategy was developed in MEDLINE and then adapted to subsequent databases. Systematic reviews, clinical-trial databases, and the grey literature were also searched (Table 1). A comprehensive sampling strategy was employed, as we expected few studies exploring this topic. The elements of STARLITE are reported (Table 1) [25]. This process is consistent with Phase 2 of Noblitt and Hare’s methodology, “Deciding what is relevant” [18].

### 2.2. Screening

Search results were transferred into Endnote, duplicates were removed, and then transferred to Rayyan (https://www.rayyan.ai, accessed on 29 March 2022) for screening. Irrelevant results were excluded based on the title. The titles and abstracts of remaining records were then screened independently by two reviewers (LNS and CVH) retaining potentially eligible studies for full-text screening. Reasons for exclusion were recorded. Disagreements at either screening stage were discussed and resolved by consensus or involvement of a third reviewer (FW). Interrater reliability was assessed at both stages by using the kappa statistic [26]. Screening and discussion were conducted in deciles and continued until there was near-perfect agreement. 

### 2.3. Quality Assessment

Quality assessment was performed independently by two reviewers (LNS and CVH), using the CASP qualitative checklist [27]. Disagreements were resolved via discussion. 

### 2.4. Data Extraction (Selection and Coding)

Consistent with Noblitt and Hare’s Phase 3 “Reading the studies”, two reviewers (LNS and CVH) extracted data independently from included studies [18]. Data-extraction forms included contextual information to aid study translation, and both 1st-order (participant quotes) and 2nd-order constructs (study author interpretations) were extracted. Extracted data were then compared to identify errors or discrepancies. 

Phase 4, according to Noblitt and Hare, involves “Determining how studies are related” by creating “a list of the key metaphors, phrases, ideas, and/or concepts” [18]. Included studies were imported into QSR NVivo (released March 2020) and coded to develop initial themes and then compared across studies to identify similarities. Initial themes were iteratively developed by two reviewers (LNS and CVH). Themes were grouped into overarching concepts and further refined through discussion with the entire review team, using concept maps created using Google Jamboard (https://jamboard.google.com accessed on 31 May 2022). The above process was performed separately for HCPs and women with rUTIs. 

### 2.5. Translation and Synthesis

The translation of the studies was conducted separately for patients and HCPs by two reviewers (LNS and CVH), as described by Noblitt and Hare’s Phase 5, “Translation of studies” [18]. We performed reciprocal translation rather than refutational, as interpretations were similar across studies. During reciprocal translation, we considered key contextual information, such as the age of study participants and study setting. As the order in which studies are translated can influence the synthesis, we started with “key papers” we considered conceptually rich [19,28]. A table was developed as described by Britten et al. to aid translation, consisting of the overarching concepts and our interpretations of 1st- and 2nd-order constructs [20]. During Phases 6 (“Synthesising translations”) and 7 “Expressing the synthesis”, this table was used to develop 3rd-order constructs iteratively [18]. The tables also allowed wider discussion with the review team and the study Patient and Public Involvement (PPI) team to consider alternative interpretations and develop the final 3rd-order constructs. Since studies discussed different aspects of rUTIs, a “line of argument” synthesis was conducted, and conceptual models were iteratively developed for both HCPs and patients. Confidence was assessed in each review finding, using the GRADE-CERQual Interactive Summary of Qualitative Findings (iSoQ) online tool [29,30].

### 2.6. Reviewers’ Reflexivity 

The review team included General Practitioners (GPs) (LNS, HA, and AE), a qualitative methodologist (FW), a statistician (RCJ), social researchers experienced in qualitative research methods (CVH and DL), an information specialist (AW), and lay contributors who were women with lived experience. The multidisciplinary nature of the team supported collective thinking and removal of professional bias. We also considered how the lead researcher, as a male GP, could impact interpretations and mitigated this by maintaining a reflexive stance throughout, conducting the synthesis jointly with a female non-clinician, undertaking regular discussion with the wider review team and involvement of the PPI team throughout. 

## 3. Results

Ten primary studies were deemed to be eligible for inclusion (Figure 1) with an overall kappa statistic of 0.71 (substantial agreement). Two additional studies were identified in an updated search in June 2022. Nine studies were conducted in Europe, two in the USA, and one in both (Table 2). Data were mainly collected by semi-structured interviews or focus groups. Two studies analysed online data involving 88,975 online posts [5,31]. Total study populations (not including online studies) were 184 patients with UTIs and 146 HCPs (93 were GPs or GP Trainees). Three studies were deemed high quality, five moderate, and three low quality (Table 2).

Confidence in the review findings are presented in Table 3. Conceptual categories and interpretations of both 1st-order and 2nd-order constructs are presented in Table 4 and Table 5 alongside developed 3rd-order constructs.

### 3.1. GRADE-CERQual (Confidence in the Evidence from Reviews of Qualitative Research

There was moderate confidence for the review findings for women with rUTIs mainly due to minor-to-moderate concerns for study methodology, adequacy, and relevance (Table 3). For HCPs, our assessment of confidence in the evidence for each review finding was low mainly due to serious concerns about the adequacy of the data (i.e., limited studies and/or thin data) (Table 3). 

### 3.2. Women with rUTIs

1.Most women describe a long history of rutis, with variability of frequency, severity, symptoms, and ability to self-diagnose

Women described a history of up to “several decades” [5]; however, there is variation in the severity and frequency of rUTIs experienced. Some described their UTIs starting in childhood or early adulthood, whilst others said that the onset related to the menopause. Many stated that the frequency and severity of symptoms worsened with time. 

*“Frequency and severity of symptoms also vary considerably with some women reporting ‘a serious attack about once every two or three months’ (Beth) compared to others who suffer from ‘constant pain’” **(1st and 2nd order FLOWER 2014**
*[5]***).***

The symptoms reported were diverse and wide ranging. Typical symptoms included dysuria, frequency, fever, and abdominal and back pain. However, several atypical symptoms were also described, including dizziness, hand and leg symptoms, and itching, amongst others. Women also described systematic symptoms, including feeling generally unwell. 

*“The women in this study described a wide range of physical health problems, including typical and atypical symptoms of UTI. The women felt their general health was affected, their body felt abnormal, and they felt tired and dull and experienced a general feeling of weakness. The participants said that they felt bad all over. Some women expressed more atypical symptoms such as dizziness and pain in the hands” **(2nd order ERIKSSON 2014**
*[34]***).***

Many women became “experts” at self-diagnosing UTI recurrence, and this led some to question the need for diagnostic tests to confirm a UTI. However, the ability to self-diagnose was not universal, with some struggling to identify recurrence. 

2.Various factors are described as triggers for ruti, and patients worry about a serious underlying cause

Sexual intercourse was commonly reported, as were hormonal changes associated with the menopause and menstruation, the use of contraception, and personal hygiene. The risk factors described included the presence of dementia, diabetes, genital prolapse, and vaginal atrophy. Additionally, some women described stress as a trigger. Women appeared to search for personal factors linked with their recurrent infections, and some blamed themselves for subsequent infections or treatment failures. Environmental factors were also a perceived cause. 

*“Participants became anxious about suspected triggers of uUTIs [uncomplicated UTI], such as a new sexual partner, restarting birth control pills, giving birth, stressful life experiences, and even cool weather… some women blamed themselves for treatment failure and recurrent uUTIs” **(2nd order GRIGORYAN 2022**
*[35]***).***

Women with an rUTI worry about an underlying cause for their symptoms, especially if they have a history of cancer. 

*“Respondents felt that there had to be something wrong with their bodies to be getting so many UTIs. ‘Why do you get a cystitis every time? There has to be something wrong somewhere’” **(1st and 2nd order PAT 2022**
*[23]***).***

Some women were also worried about long-term and persistent urinary symptoms.

3.UTIs generally have significant and widespread effects on women’s quality-of-life

Women described significant and widespread negative effects on their lives. Frequency of urination and urgency were particularly disruptive. They described significant disruptions to sleep due to nocturia, resulting in tiredness and fatigue. Some women also described new or worsening urinary incontinence, resulting in anxiety, fear and “the need to be close to a bathroom” [35].

*“It is apparent that rUTIs have a far greater impact on the quality of their lives than is commonly acknowledged in the medical literature. Even relatively mild symptoms of frequency and urgency can disrupt sleep, create anxiety, and lead to persistent fatigue” **(2nd order FLOWER 2014**
*[5]***).***

Women reported several negative psychological impacts of rUTIs: feeling depressed, anxious, and irritable; having reduced concentration; not feeling clean; and a fear of giving off a bad odour. Women also described feeling helpless and frustrated due to recurrent infections and blaming themselves for repeated infections. After acute-infection resolution, women worried about future UTIs and the impact they could have on future plans. Finally, needing to seek healthcare itself resulted in anxiety and frustration.

*“The women thought their mood was affected and they felt depressed and dejected. They said living with repeated UTI is a misery ... ‘I usually say that it’s a misery because that is really what it is’” **(1st and 2nd order ERIKSSON 2014**
*[34]***).***

Some experienced *“dread and anxiety because of the expectation of another uUTI, as well as anticipation of the negative symptoms to come” **(1st and 2nd order GRIGORYAN 2022**
*[35]***).***

Recurrent UTIs had a significant impact on relationships for most women. During infections, women often cancelled plans with friends, often becoming lonely and isolated. They also reported significant long-term impacts on intimate relationships. During acute infections, women feared incontinence, were concerned about odour, and reported that sex was the “last thing on their minds” [38]. As sexual intercourse is a risk factor for UTIs, women also reported a fear of sex, even when infection free, significantly impacting their personal relationships.

*“Some of the women said that sexual activities were affected when they had a UTI. They had less sexual desire partly from fear that they would have urine leakage and smell bad. When they had a UTI, they avoided sexual activities: ‘You don’t want to have sexual intercourse when you don’t feel well and it hurts and there’s a risk you’ll have to go to the toilet. Besides, it feels unhygienic’” **(1st and 2nd order ERIKSSON 2014**
*[34]***).***

Recurrent UTIs also significantly impacted work, education, childcare, and home life, with symptoms resulting in reduced productivity and further feelings of frustration. Infections often led to time off work due to symptoms and/or seeking healthcare, with potential negative financial impacts. In countries where healthcare was privately funded, the financial impact was even greater. Enjoyment and interest in hobbies was also impacted negatively by rUTIs. 

*“RUTIs can also disrupt other areas of a woman’s life with many reports of enforced periods off work with financial and social consequences: ‘The UTIs have taken a huge toll on my sex-life (and therefore my relationship), work, social life and finances’” **(1st and 2nd order FLOWER 2014**
*[5]***).***

The severe negative impacts of rUTIs were not universal. Some described being “used” [34] to having UTIs, whilst others did not describe significant impacts of their rUTI on their lives.

*“However, there were women in this study who did not report any effects on their social or sexual activities. In addition, 26 women declined to participate and some of them said they did not find UTIs especially bothersome” **(2nd order ERIKSSON 2014**
*[34]***).***

4.Women with rutis commonly use self-help, lifestyle, and complementary and alternative medicine (cam) options to try to treat and prevent UTIs.

Several women described numerous lifestyle and CAM methods to both treat and prevent further UTIs. Complementary treatments were described as having varying success, but there was significant interest in their use. Several different types of CAM were described, from herbal medicines to acupuncture and reflexology. Some used them as an alternative to antibiotics, whereas others reported using them in addition to antibiotics, for example, to reduce antibiotic side-effects. 

*“Women on the forum show considerable interest in other conventional treatments and in Complementary and Alternative Medicines (CAM) including dietary and lifestyle changes and acupuncture and herbal medicines. Typically women may use several different therapeutic modes that are combined into a complex CAM intervention” **(1st and 2nd order FLOWER 2014**
*[5]***).***

Women reported employing numerous lifestyle measures to prevent and treat UTIs. The most reported method was increasing fluid intake, but other lifestyle changes included post-coital voiding, changes to personal hygiene, rest, and keeping warm. In some cases, this was related to what women believed was the underlying causes for their UTIs. Over-the-counter medications were also frequently described, including vitamins, d-mannose, and cranberry-based supplements. It appears that self-management, when successful, empowered women by providing some control over their UTIs.

*“All women started doing something themselves when they had a urinary tract infection, especially drinking a lot, taking vitamin C or drinking cranberry juice” **(2nd order GROEN 2005**
*[36]***).***

*“A few participants suffering from recurrent uUTIs mentioned feelings of positivity when they found ways of keeping the problem controlled. A participant said, ‘I did recover faster because I was drinking more water, so now I feel like I’ve got a little more empowerment every day to prevent them’” **(1st and 2nd order GRIGORYAN 2022***[35]***).***

5.The effectiveness of antibiotics varied, and patients are concerned about their use

Most women described antibiotics as effective and that long-term antibiotics can be “transformative” [5]. Based on previous positive experiences, some patients expected antibiotics with subsequent infections. However, often women with rUTIs described short-lived effects of antibiotics whilst others described minimal or no benefit. Views on antibiotics varied, from avoiding them for CAMs to those who felt they were needed to prevent worsening infection. 

*“Antibiotics emerged as trusted medicines that had widespread use. The descriptors used by the interviewed patients ranged from ‘magic answer’ to ‘sledgehammer’ treatment. Those who referred to antibiotics as a ‘magic’ pill were often older females with recurrent UTIs and with expectations for apparently appropriate prescribing” **(1st and 2nd order GULLIFORD 2021**
*[37]***).***

There was generally fear and concern about the use of antibiotics, even if their symptoms respond to them, particularly related to antimicrobial resistance (AMR), with several reporting personal experiences of AMR. Depending on discussions with their HCPs, AMR and treatment failure caused some women to blame themselves for treatment failure.

*“The first theme addressing the negative impact of antibiotics discussed by participants was the development of antibiotic resistance. During these discussions, women noted concern for their own ability to be treated for future infections, but also acknowledged the broader implications for society as a whole” **(2nd order SCOTT 2021**
*[24]***).***

*“When the physician offered no explanation for the treatment failure, participants sometimes felt responsible and blamed themselves. In the words of one woman, ‘I was asking myself whether I did something wrong: sitting on a cold surface, not drinking enough…’” **(1st and 2nd order GRIGORYAN 2022**
*[35]***).***

Antibiotic side-effects were another concern for many, including thrush and gastrointestinal side-effects. The “collateral damage” [24] of antibiotics was described, as well as worries about the impact antibiotics had on their immune system and that repeated courses increased their susceptibility to subsequent infections. Such was the level of concern of some that they resisted taking antibiotics even when prescribed.

*“More long-term concerns centre around the potential impact of antibiotics on the immune system: ‘I spent months on antibiotics, which I think knocked my immune system even more, and made it easier for the next infection to come along!’... Thus some women perceived a negative cycle in which an infection triggers antibiotics, which provide temporary relief but then make them more likely to contract another infection” **(1st and 2nd order FLOWER 2014**
*[5]***).***

6.Women with rUTIs seek healthcare for most, but not all, UTIs. they describe anger and frustration with healthcare in terms of their care, the use of antibiotics, and an underestimation of the impact of rUTIs

It appeared that women generally do not seek healthcare at the onset of symptoms for several reasons, including inconvenience and cost of attending healthcare and wanting to self-manage and avoid antibiotics. Common reasons for then seeking healthcare included persistent or worsening symptoms or a patient-performed positive urine dipstick. Some women also reported reluctance to seek healthcare and feeling guilty over seeking it.

*“Common triggers that ultimately prompted participants to seek help were symptoms that did not go away after a few days, pain that worsened to the point of being unbearable, or actively taking a test strip that came back positive” **(2nd order GRIGORYAN 2022**
*[35]***).***

When consulting for an rUTI, women felt that they were more involved in management discussions. Some patients also discussed that the goal of treatment was not necessarily a cure but decreasing the number of UTIs they experienced. 

Experiences of healthcare interactions varied, with negative experiences often being reported. Most findings were based on treatment and information experiences, investigation of their rUTIs, and their interaction with their HCP. When a patient’s expectation for information on treatment and prevention was not met, the interaction was seen negatively; further investigation was viewed positively, and patients were relieved. Others reported having to “repeatedly” [23] ask for referral for investigation. This contrasted with acute investigation, where some women questioned the need since they perceived “the diagnosis to be obvious” [35].

*“I asked to be referred to a urologist again and he said ‘Why do you want to see one? They will only put you on a preventative dose of antibiotics’. . . ‘I had to practically beg him until he finally agreed to refer me’” **(1st order FLOWER 2014**
*[5]***).***

HCPs who were supportive and knowledgeable about rUTIs and their impact were valued by women with rUTIs. In contrast, negative experiences were described when HCPs were perceived as “unsympathetic” and “dismissive” [5]. Distrust and frustration were reported by some when they did not feel listened to in terms of their concerns about antibiotic use and the impact of their rUTI; moreover, some reported anger if their HCP did not discuss non-antibiotic treatments. The impact of rUTIs was felt by some women to be underestimated by HCPs, leading to further mistrust, and this was exacerbated by gender and age. 

*“Reports of a positive interaction with GPs repeatedly emphasise a woman’s relief (and often surprise) to find that their doctor listened and was responsive to their complaint, that they had read the notes and were informed about the particular presentation of the woman and RUTIs in general. They demonstrated understanding and kindness and were willing to refer to more specialist expertise” **(1st and 2nd order FLOWER 2014**
*[5]***).***

*“When discussing their experience with prevention and treatment options for rUTIs, many participants reported resentment toward medical providers, particularly overprescribing antibiotics… They described frustration when physicians did not validate their experiences with rUTIs and fears over antibiotic use, and resentment when they were not offered nonantibiotic options for management, leading to mistrust of their providers” **(2nd order SCOTT 2021**
*[24]***).***

7.Women sought information and support from a variety of sources and feel that more information and research is needed

Many women wanted more information about the causes, treatment, and preventive options for their rUTI; the exact information needed varied between individuals. Some women reported “distress” [24] that further information and research was needed on the underlying causes and the microbiome. They also felt research on non-antibiotic treatment options was needed. A number reported receiving self-help advice from their HCP, such as hydration, post-coital voiding, and cranberry use. Most women appeared to seek extra information from a variety of sources, but most commonly from the Internet. 

*“There were also differences in the information wanted by respondents. Some only wanted information about the causes, some only wanted information about the therapeutic options, and others wanted both” **(2nd order PAT 2020**
*[23]***).***

Often, women described being unable to discuss their rUTIs with relatives or partners. It appeared that online support groups provide women with support, which ranged from discussing symptoms, risk factors, treatment, and preventative options to providing support and reassurance. 

*“‘I’m losing a lonely battle with the medical community as these UTIs won’t go away. I’m glad you [the online community] are here’” **(1st order GONZALEZ 2022**
*[31]***).***

### 3.3. Healthcare Professionals

There are differences in the use of, and reason for, conducting urine cultures

The predominate reason for conducting a urine culture was to confirm infection. Secondary reasons were identifying unusual organisms and AMR. HCPs highlighted several limitations of urine sampling, including cost, concerns about over-medicalising UTIs, and that urine culture does not impact initial UTI management.


*“Doctors are more likely to request MSUs in recurrent UTI as they want proof of infection for any future referral, as well as confirming the diagnosis. ‘If somebody comes with a repeat, just to make sure it is not an irritable bladder I would send an MSU’” **(1st and 2nd order, LARCOMBE 2012** [38]**).***


Most HCPs appeared to use urine culture in this population. However, some used it less since they believed their patients were capable of self-diagnosing due to previous experience.


*“A few investigate less in recurrent UTI for the logical reason that women are well versed in their symptoms and diagnostic confirmation isn’t usually necessary” **(2nd order, LARCOMBE 2012** [38]**).***


2.Making the correct diagnosis is important, and HCPs worry about missing serious disease

HCPs considered several potential causes of rUTIs, including lifestyle factors related to sexual intercourse, hygiene, and hydration. They also considered medical and serious causes, such as cancer, diabetes, or pyelonephritis, and felt it important to make the correct diagnosis. Due to concerns of pyelonephritis, HCPs generally would assess these patients in person and not remotely. 


*“GPs also emphasised the importance of making the correct diagnosis for a UTI and not overlooking more serious pathology that can present with similar symptoms such as diabetes, bladder cancer, or pyelonephritis” … “Examples of lifestyle factors perceived relevant by GPs, of which sexual intercourse was highlighted as particularly important” **(2nd order, FLOWER 2015** [22]**).***


3.HCPs appreciate that rUTIs have a significant impact on quality-of-life

HCPs considered rUTIs to have a significant impact on patients, and one GP described them as “quality of life wrecking” [22]. They understood there can be widespread impacts on patients, both physically and psychologically. Symptoms that were described as particularly impactful were urine frequency and nocturia. In terms of the psychological impact, HCPs believed that rUTIs can cause patients to be “fearful”, anxious, and “depressed” [22]. HCPs also appreciated the detrimental effects on patients’ personal relationships, ability to work, and social lives. 


*“I think it has a huge impact on their lives, because … some of them … because they are experiencing such symptoms of frequency … um … they won’t actually go out of the house very often … the say they’re scared to leave the house … interrupts their sleep of course, they’re up numerous times through the night …um… they’re constantly on edge … they get very tired …um … and they can get quite depressed ‘… it can interfere with their sexual life and a lot of them feel dirty—and a lot of them are reluctant to embark on sexual relationships because of the risk of a recurrent UTI” **(1st order, FLOWER 2015** [22]**).***


The impact of rUTIs on patients appeared to influence HCPs’ antibiotic decision-making, and they were more open to immediate antibiotic prescribing. The provision of delayed antibiotics (antibiotics not used immediately but in a few days if symptoms worsen [40]) was also seen as a method to empower patients and give them some control.


*“Participants also saw the provision of delayed prescriptions for antibiotics as a way of enabling women to ‘get on top of their symptoms and control the amount of impact it has on their lives’ (GP14) which can ‘empower’ them and restore their sense of ‘hope’” **(1st and 2nd order, FLOWER 2015** [22]**).***


4.HCPs believe self-help measures are important to prevent UTI recurrence

Most HCPs considered self-management important. Some felt UTIs were overtreated, and more advice given to patients would improve self-care. Examples of self-care that HCPs promote include hydration and personal hygiene methods. However, there was a suggestion that discussing self-management with patients with rUTIs may have limited value, as they generally have already been using these methods before presentation. 


*“Others felt that infections were ‘over-treated’ (GP12) and that more information and ‘self-help advice’ (GP11) such as drinking more water, reducing caffeinated drinks (GP5) or using over-the-counter remedies such as cranberry (GP15) could improve self-management” **(1st and 2nd order, FLOWER 2015** [22]**).***


5.Some HCPs are interested in the use of CAMs but have significant concerns about their use and admit a lack of knowledge about them

Some HCPs were interested in the use of CAMs, especially in cases where conventional treatments had failed. They felt that CAMs could be effective and are “biologically plausible” [22], but some HCPs had significant reservations. Concerns, varying from minimal to very concerned, related to effectiveness, safety, and the quality assurance and regulation of herbal practitioners. Additional concerns focussed on the cost to patients, the impact that negative effectiveness could have on their relationship with their patients, and the potential legal risk of recommending such treatments. 


*“In particular, concerns were expressed about the lack of quality assurance of herbal products, … and possible interactions between herbs and drugs. The training, lack of regulation, or licensing, and the level of medical knowledge of herbal practitioners were additional sources of GP uncertainty” **(2nd order, FLOWER 2015** [22]**).***


Based on the above concerns and most HCPs describing poor knowledge of CAMs, they were reluctant to recommend them. However, if evidence of effectiveness and safety were established and concerns regarding regulation and quality assurance were resolved, HCPs would be more open to recommending them.

6.HCPs feel antibiotics are effective short term but are concerned about their use

HCPs described significant concerns about antibiotic use for rUTIs due to its “recalcitrant” and “chronic nature” [22]. Concerns related to the development of AMR and side-effects. In the context of AMR, most HCPs stated that they would discuss this concern with patients.


*“Most groups reported that they would discuss antibiotic resistance with some patients with an expected UTI, especially if there were additional risk factors” (recurrent UTIs) **(2nd order COOPER 2020** [39]**).***


7.Attitudes toward when to refer to secondary care and its benefits varied amongst HCPs

Some HCPs used a specific number of UTIs to decide about referral; others referred if the UTI were “not controlled”, and others used “gut feeling” [22]. Experiences included that *“some GPs regarded referrals as a ‘waste of time’ and thought they resulted in unnecessary procedures such as cystoscopies that had little impact on management of the condition ‘because in the vast majority of them, they don’t find any significant abnormalities’”* [22]. 

8.HCPs must consider and balance several competing interests when managing rUTIs

HCPs described managing several competing interests, including improving symptoms, avoiding over-medicalisation, AMR concerns, lacking guidance, and patient expectation. There is a desire to improve patients’ symptoms resulting in emotions such as helplessness with unsuccessful outcomes. Some HCPs felt that UTIs were overtreated and giving antibiotics for a “self-limiting condition re-enforced the dependency on their GP” [22]. Moreover, guidance advises reduced antibiotic prescribing due to AMR concerns. Patient expectation also appeared to be an important factor considered by HCPs. Many felt that patients’ expectations were increasing and that patients wanted to know the underlying cause of their rUTIs, which some found difficult to answer, and had an increased expectation for antibiotics. Finally, HCPs stated a need for further guidance on rUTIs due to their increased perceived complexity.


*“Many professionals think that patient expectations and in particular, the need for explanations, were rising. Increased expectation concerns professionals when they can’t meet these needs” **(2nd order, LARCOMBE 2012** [38]**).***


### 3.4. Conceptual Model Development

Based on the findings above, two conceptual models were developed, one for the patient and the other for the HCP (Figure 2 and Figure 3).

## 4. Discussion

This is the first qualitative evidence synthesis to present the experiences and views of both women with rUTIs and HCPs managing them. Women with rUTIs have a unique experience, with varying symptoms, frequency of recurrences, severity, and impacts on their lives. Several triggers for rUTIs were described, including sexual intercourse, hormonal changes around the time of menstruation and menopause, the use of contraception, or personal hygiene. They value the short-term effects of antibiotics but highlight their limitations and have concerns about their use. They often use and are interested in non-antibiotic treatments and want more information on these and the underlying cause of their rUTIs. Views of healthcare varied depending on whether they felt listened to and supported by their HCP. There is also frustration and anger about antibiotic use, with limited discussion of alternative treatments. Based on the GRADE-CERQual assessment and triangulation with our PPI team, we have moderate confidence in these findings. 

HCPs report that they appreciate the impact rUTIs have on patients and want to improve their symptoms. Recurrent UTIs are complex and challenging to manage, from making the diagnosis and not missing a serious cause to managing the competing interests of patient expectations, concerns, and guidance about antibiotic use and whether to refer for further investigation. How HCPs manage treatment failure varies, and it is in this context that they would consider CAM, although concerns about safety limit recommendations for CAMs. Based on the GRADE-CERQual assessment, we have low confidence in these findings. 

There was general agreement between patients and HCPs in terms of the causes of rUTIs, concern for an underlying cause, the impact they have, antibiotic effectiveness and concerns, and self-management options. There were some conflicts suggesting possible communication gaps, as patients perceived that HCPs underestimated the impact of rUTIs and in terms of the expectation for antibiotics for acute UTIs. HCPs varied in their views on referral for further investigation, whereas patients viewed HCPs negatively if they were not referred.

### 4.1. Comparison with the Existing Literature

The literature on the experiences of patients with rUTIs or UTIs in general is very limited, with only one other UTI-related QES [7]. There are several parallels between our findings and theirs. Both reviews found that the symptoms of rUTIs are diverse and often severe and associated with systemic symptoms. We also found that these symptoms have significant psychological, emotional, and social impacts on patients, with similarities to those seen in women with urinary incontinence [41]. However, in addition, we found variability between women, with some reporting minimal-to-no significant impacts on their lives. We also describe significant impacts on intimate relationships due to concerns of hygiene, and this finding has also been seen in urinary incontinence [41]. This is not surprising since some of our findings appear to be mainly due to new or worsening incontinence. However, since intercourse is a recognised risk factor for UTIs, this also impacted intimate relationships after acute infection resolution. Both our findings and those of Izett-Key et al. found that patients were worried about an underlying cause, found antibiotics effective, and had concerns about AMR [7]. This concern about AMR appears to contradict previous research suggesting low patient awareness [42]. There was also overlap in terms of patients wanting more information about their condition, and our study adds further knowledge in terms of where patients access this information and the support networks they access online. 

There are currently no QES on the experiences and concerns of HCPs in the context of UTIs. However, some of the findings from this review have been seen in other studies on UTIs [43,44,45,46,47,48]. The findings from this review suggest that HCPs are concerned about the development of AMR, and this is one of the competing interests they must consider when managing patients with rUTIs. This agrees with other qualitative work suggesting HCP concerns about AMR, and it appears that this concern is increasing since previous work in Sweden suggested that some GPs did not have any concerns at all about AMR and had not come across it in their practice [43,44,45]. HCPs also perceived that their patients expected to be prescribed antibiotics. This belief has been shown in other qualitative work in UTIs and respiratory tract infections [44,46,47,48]. However, as suggested from the patient perspective and other studies, this perceived expectation for antibiotics may not be correct in all cases and may be decreasing with time [44,48,49]. 

### 4.2. Strengths and Limitations

This qualitative evidence synthesis has several strengths. The search conducted was comprehensive and involved several strategies to identify relevant studies and did not restrict on language or date. This review also addresses an important evidence gap in terms of the experiences of women with rUTIs and HCPs. The QES was performed by a multidisciplinary team, with attention paid to reflexivity. Finally, we appraised our confidence in the review findings, using GRADE-CERQual, as recommended by Cochrane [29].

This review does have some limitations. For some studies, it was difficult to disentangle whether findings were specifically related to women with rUTIs. We undertook several strategies to avoid including data not related to women with rUTIs, but potentially relevant data may have been excluded. In addition, during the screening process, relevant studies may have been excluded, as they had not explicitly mentioned including women with rUTIs. Finally, there were limited data for HCPs in this area, and this is reflected in our low confidence in these review findings. Only one study focused on rUTIs, and one of its main aims was to explore attitudes toward herbal medicines, thus potentially biasing the results.

### 4.3. Clinical Implications and Future Research

Based on the GRADE-CERQual assessment, the current literature does not fully answer the questions of HCPs’ views on antibiotic-alternative treatments and, in this context, what specific information patients want. Further qualitative work is required to adequately answer these questions. One method of providing this information could be in a treatment-option decision aid to support SDM. It would also ensure that the information patients receive is accurate, based on current knowledge, and avoids patients seeking information from other sources, which could be misleading, incorrect, and harmful. We aim to address these evidence gaps via further qualitative work and develop a patient-decision aid for primary care (IMproving Prophylactic Antibiotic use for Recurrent urinary Tract infection (IMPART): mixed-methods study to address evidence gaps and develop a decision aid (NIHR-FS-2021-LS)). 

Additional explicit guidance on when referral should be considered and the rationale for urine culture in rUTIs would be beneficial. Future research focused on if and how the views and experiences of women with rUTIs differ based on age and menopausal status would also be beneficial. HCPs were concerned about AMR in the context of rUTIs; however, their views on its consequences were not reported, and it is another area that warrants further research. Finally, it appears that there is a communication gap between patients and HCPs on the impact of rUTIs, the expectation for antibiotics, and treatment failure. This review suggests that it is important that HCPs acknowledge the impact of rUTIs and explore patient expectations of the consultation to avoid these misunderstandings. It is also important that, in the context of treatment failure, this is discussed within the consultation to avoid patients blaming themselves.

## 5. Conclusions

This qualitative evidence review demonstrates that the experience of women with rUTIs is unique but is generally of a chronic condition with significant impacts on numerous aspects of their lives. Antibiotics can be “transformative”, but patients have serious concerns about their use and feel that non-antibiotic options need further research and discussion. HCPs share similar views about the impacts of rUTIs and concerns about antibiotic use and find the management of rUTIs to be complex and challenging. Further development of current guidance and the development of a patient facing decision aid could help address some of the gaps highlighted in this review.

## Figures and Tables

**Figure 1 antibiotics-12-00434-f001:**
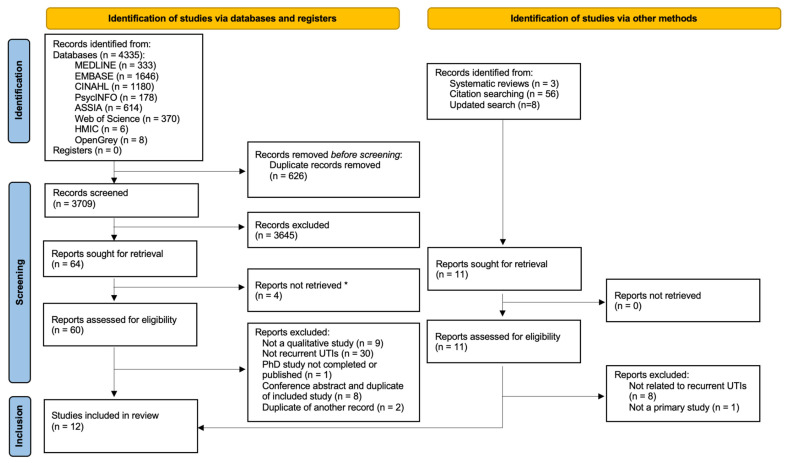
PRISMA flow diagram of results of database search of the literature [32]. * Articles could not be retrieved despite attempts to contact authors or journals.

**Figure 2 antibiotics-12-00434-f002:**
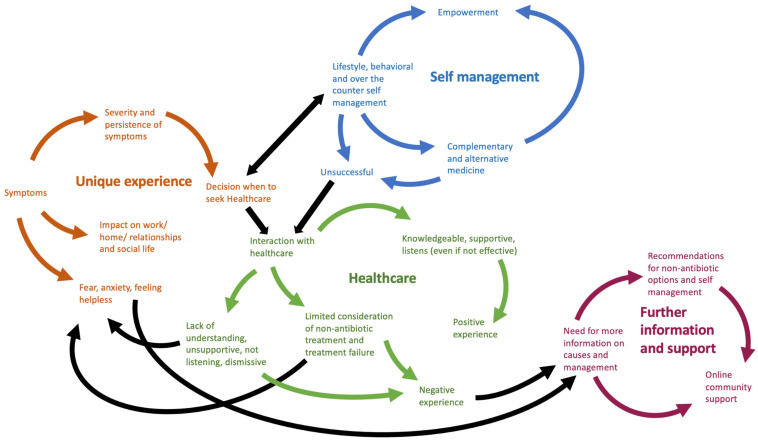
Conceptual model of patients with rUTIs.

**Figure 3 antibiotics-12-00434-f003:**
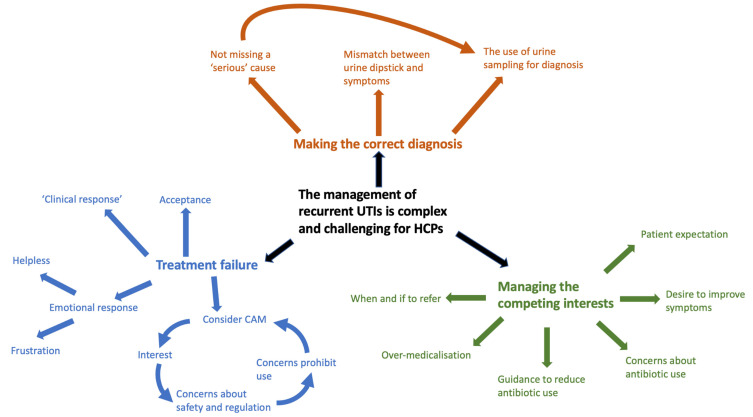
Conceptual model for healthcare professionals.UTI = urinary tract infection; HCP= healthcare professionals; CAM = Complementary and Alternative Medicine.

**Table 1 antibiotics-12-00434-t001:** Elements of STARLITE for the search strategy [25].

STARLITE Category	Description
Sampling strategy	Comprehensive.
Type of studies	Primary qualitative studies using both qualitative methods for data collection and analysis.
Approaches	Electronic databases, reference-list searching, and citation tracking of included studies.
Range of years	Primary search—database conception to November 2021. Updated search—November 2021 to June 2022.
Limits	No limits.
Inclusion and exclusions	Primary care/General Practice or outpatient secondary care. Inclusion criteria: Women aged 18 years and older with rUTI. HCPs managing adult women with rUTI. Primary qualitative studies and mixed methods studies using qualitative methods to collect data and analyse them. Exclusion criteria: Patients requiring intermittent self-catheterisation or long-term catheterisation. Patients who have undergone pelvic or bladder surgery or radiotherapy. Patients with multiple sclerosis or another neurological condition affecting the urinary tract. Pregnant patients.
Terms used	MEDLINE search strategy: exp urinary tract infections/ exp cystitis/ exp pyelitis/ (UTI or RUTI or cystitis* or bacteriuria* or pyelonephriti* or pyonephrosi* or pyelocystiti* or pyuri* or urosepsis* or uroseptic*).ti,ab. ((bladder* or genitourin* or genito urin* or kidney* or pyelo* or renal* or ureter* or ureth* or urin* or urolog* or urogen* or urinary tract*) adj3 (infect* or bacteria* or microbial* or sepsis* or nflame*)).ti,ab. 1 or 2 or 3 or 4 or 5 Gender Identity/ Sex Characteristics/ Sex Determination/ Sex Distribution/ Sex Factors/ exp Women/ Womens Health/ Womens Health Services/ Health Care for Women International.jn. Journal of The American Medical Womens Association.jn. Women & Health.jn. Womens Health Issues.jn. female*.tw. gender*.tw. girl*.tw. mother*.tw widow*.tw. woman*.tw. women*.tw. or/7–25 ((“semi-structured” or semistructured or unstructured or informal or “in-depth” or indepth or “face-to-face” or structured or guide) adj3 (interview* or discussion* or questionnaire*)).ti,ab. (focus group* or qualitative or ethnograph* or fieldwork or “field work” or “key informant”).ti,ab. interviews as topic/ or focus groups/ or narration/ or qualitative research/ 27 or 28 or 29 6 and 26 and 30
Electronic sources	MEDLINE, Embase, CINAHL, PsycInfo, ASSIA, Web of Science, Cochrane Database of Systematic Reviews, Epistemonikos, Cochrane Central Registry of Controlled Trials, OpenGrey, and the Health Management Information Consortium (HMIC).

STARLITE = Sampling strategy, Type of study, Approaches, Range of years, Limits, Inclusion and exclusions, Terms used, and Electronic sources.

**Table 2 antibiotics-12-00434-t002:** Characteristics of included patients in healthcare professional studies (ordered by study, first author).

Reference (Ref. no.) and Country	Population/Setting	Method of Data Collection	Method of Data Analysis	Study Participants	CASP Quality Appraisal Rating
Croghan 2021 [33] Ireland	Women recruited from secondary-care clinics (aged 23–38 years)	Semi-structured interviews	Thematic analysis	10 participants	Low
Eriksson 2014 [34] Sweden	Women aged 67–96 years from primary care	Semi-structured interviews	Qualitative content analysis	20 participants	High
Flower 2014 [5] U.K.	Postings on a web forum (women who disclosed age were 13–65 years)	“Qualitative description”	Qualitative description	5386 postings (915 topics)	Moderate
Gonzalez 2022 [31] U.S.A.	Online postings (ages not stated)	“Data mining”	Qualitative thematic analysis and a latent Dirichlet allocation	83,589 online posts by 53,460 users (from 859 websites)	Low
Grigoryan 2022 [35] Germany and U.S.A.	Women aged 18 to over 70 years recruited via a patient community panel and physician referrals (50% had experienced rUTI)	Semi-structured interviews with semi-structured narrative self-task component	Thematic synthesis	65 participants (35 participants with > 2 UTIs in the last year)	Moderate
Groen 2005 [36] The Netherlands	Women aged 15–49 (15 were aged over 20 years) from primary care	Semi-structured interviews	Not stated	21 participants	Low
Gulliford 2021 [37] U.K.	Women and men aged 20–99 years attending GP with an infection	Semi-structured interviews	Thematic synthesis	31 participants (11 with a UTI)	Moderate
Larcombe 2012 [38] U.K.	Women aged 18–64 years from primary care (rUTI and non-rUTI)	Focus groups and semi- structured interviews	Grounded theory	2 focus groups (10 participants) and 12 interviews	High
Pat 2020 [23] The Netherlands	Women referred to secondary care (Urology Department) (aged 32–86 years)	Semi-structured interviews	Thematic analysis	6 participants	Moderate
Scott 2021 [24] U.S.A.	Women aged 20–81 years recruited from “female pelvic medicine and reconstructive surgery clinics”	Focus groups	Grounded theory	6 focus groups, 29 participants	Moderate
Cooper 2020 [39] U.K.	Primary care staff (ages not stated)	Focus groups	Theoretical domains framework	8 focus groups, 57 participants	Moderate
Flower 2015 [22] U.K.	GPs (aged 34–59 years)	Semi-structured interviews	Thematic analysis	15 participants	Moderate
Larcombe 2012 [38] U.K.	Primary-care staff, including GPs, GPs in training, nurses, pharmacists, administrative staff, and medical managers (ages not stated)	Focus groups and semi-structured interviews	Grounded theory	12 focus groups (72 participants) and 2 interviews.	High

**Table 3 antibiotics-12-00434-t003:** GRADE-CERQual assessment of confidence in review findings. For the full evidence profile for the GRADE-CERQual assessment, see Appendix A). Tables were developed using the GRADE-CERQual (iSoQ) online tool [30].

	Summarised Review Finding	GRADE-CERQual Assessment of Confidence	Explanation of GRADE-CERQual Assessment	References
Patients with Recurrent UTIs				
**1**	Most women describe a long history of rUTIs, with variability of frequency, severity, symptoms, and ability to self-diagnose.	Moderate confidence	Minor concerns regarding methodological limitations, no/very minor concerns regarding coherence, minor concerns regarding adequacy, and minor concerns regarding relevance.	Grigoryan et al., 2022 [35]; Croghan et al., 2021 [33]; Scott et al., 2021 [24]; Eriksson et al., 2014 [34]; Flower et al., 2014 [5]; Larcombe 2012 [38]
**2**	Various factors are described as triggers for rUTIs, and patients worry about a serious underlying cause.	Moderate confidence	Moderate concerns regarding methodological limitations, no/very minor concerns regarding coherence, minor concerns regarding adequacy, and minor concerns regarding relevance.	Grigoryan et al., 2022 [35]; Groen and Lagro-Janssen 2005 [36]; Pat et al., 2020 [23]; Gonzalez et al., 2022 [31]; Flower et al., 2014 [5]; Larcombe 2012 [38]
**3**	Recurrent UTIs generally have significant and widespread effects on women’s quality-of-life.	Moderate confidence	Moderate concerns regarding methodological limitations, no/very minor concerns regarding coherence, minor concerns regarding adequacy, and minor concerns regarding relevance.	Grigoryan et al., 2022 [35]; Groen and Lagro-Janssen 2005 [36]; Croghan et al., 2021 [33]; Pat et al., 2020 [23]; Gonzalez et al., 2022 [31]; Eriksson et al., 2014 [34]; Flower et al., 2014 [5]; Larcombe 2012 [38]
**4**	Women with an rUTI commonly use self-help, lifestyle, and Complementary and Alternative Medicine (CAM) options to try to treat and prevent UTIs.	Moderate confidence	Moderate concerns regarding methodological limitations, no/very minor concerns regarding coherence, minor concerns regarding adequacy, and minor concerns regarding relevance.	Grigoryan et al., 2022 [35]; Groen and Lagro-Janssen 2005 [36]; Pat et al., 2020 [23]; Scott et al., 2021 [24]; Gonzalez et al., 2022 [31]; Eriksson et al., 2014 [34]; Flower et al., 2014 [5]; Larcombe 2012 [38]
**5**	The effectiveness of antibiotics varied, and patients are concerned about their use.	Moderate confidence	Minor concerns regarding methodological limitations, no/very minor concerns regarding coherence, minor concerns regarding adequacy, and minor concerns regarding relevance.	Grigoryan et al., 2022 [35]; Gulliford et al., 2021 [37]; Pat et al., 2020 [23]; Scott et al., 2021 [24]; Gonzalez et al., 2022 [31]; Flower et al., 2014 [5]; Larcombe 2012 [38]
**6**	Women with rUTIs seek healthcare for most, but not all, UTIs. They describe anger and frustration with healthcare in terms of their care, the use of antibiotics, and an underestimation of the impact of rUTIs.	Moderate confidence	Moderate concerns regarding methodological limitations, minor concerns regarding coherence, no/very minor concerns regarding adequacy, and minor concerns regarding relevance.	Grigoryan et al., 2022 [35]; Pat et al., 2020 [23]; Scott et al., 2021 [24]; Gonzalez et al., 2022 [31]; Eriksson et al., 2014 [34]; Flower et al., 2014 [5]; Larcombe 2012 [38]
**7**	Women sought information and support from a variety of sources and feel that more information and research are needed.	Moderate confidence	Moderate concerns regarding methodological limitations, no/very minor concerns regarding coherence, no/very minor concerns regarding adequacy, and minor concerns regarding relevance.	Pat et al., 2020 [23]; Scott et al., 2021 [24]; Gonzalez et al., 2022 [31]; Eriksson et al., 2014 [34]; Flower et al., 2014 [5]
Healthcare Professionals				
**8**	There are differences in the use of, and reason for, conducting a urine culture.	Low confidence	Minor concerns regarding methodological limitations, no/very minor concerns regarding coherence, serious concerns regarding adequacy, and no/very minor concerns regarding relevance.	Flower et al., 2015 [22]; Cooper et al., 2020 [39]; Larcombe 2012 [38]
**9**	Making the correct diagnosis is important, and HCPs worry about missing serious disease.	Low confidence	Minor concerns regarding methodological limitations, no/very minor concerns regarding coherence, serious concerns regarding adequacy, and no/very minor concerns regarding relevance.	Flower et al., 2015 [22]; Cooper et al., 2020 [39]
**10**	HCPs appreciate that rUTIs have a significant impact on quality-of-life.	Low confidence	Minor concerns regarding methodological limitations, no/very minor concerns regarding coherence, serious concerns regarding adequacy, and no/very minor concerns regarding relevance.	Flower et al., 2015 [22]
**11**	HCPs believe self-help measures are important to prevent UTI recurrence.	Low confidence	Minor concerns regarding methodological limitations, minor concerns regarding coherence, serious concerns regarding adequacy, and no/very minor concerns regarding relevance.	Flower et al., 2015 [22]; Cooper et al., 2020 [39]
**12**	Some HCPs are interested in the use of CAMs but have significant concerns about their use and admit a lack of knowledge about them.	Low confidence	Minor concerns regarding methodological limitations, no/very minor concerns regarding coherence, serious concerns regarding adequacy, and no/very minor concerns regarding relevance.	Flower et al., 2015 [22]
**13**	HCPs feel that antibiotics are effective in the short term but are concerned about their use.	Low confidence	Minor concerns regarding methodological limitations, no/very minor concerns regarding coherence, serious concerns regarding adequacy, and no/very minor concerns regarding relevance.	Flower et al., 2015 [22]; Cooper et al., 2020 [39]
**14**	Attitudes toward when to refer to secondary care and its benefits varied amongst HCPs.	Low confidence	Minor concerns regarding methodological limitations, no/very minor concerns regarding coherence, serious concerns regarding adequacy, and no/very minor concerns regarding relevance.	Cooper et al., 2020 [39]; Larcombe 2012 [38]
**15**	HCPs must consider and balance several competing interests when managing rUTIs.	Low confidence	Minor concerns regarding methodological limitations, no/very minor concerns regarding coherence, serious concerns regarding adequacy, and no/very minor concerns regarding relevance.	Flower et al., 2015 [22]; Cooper et al., 2020 [39]; Larcombe 2012 [38]

**Table 4 antibiotics-12-00434-t004:** Conceptual categories, 1st-, 2nd-, and 3rd-order constructs for women with rUTIs.

Concepts	1st- and 2nd-Order Construct Interpretations	3rd-Order Constructs	Number of Studies
Duration and frequency of rUTIs	Generally, a long history of rUTIs stretching back to early adulthood or childhood. Frequency of recurrences varies from “constant” to “every two or three months” [5].	Most women with an rUTI describe a long history of recurrent UTIs with variability in terms of UTI frequency, severity, presence of typical and atypical symptoms, and ability to self-diagnose recurrence.	6
Wide-ranging typical and atypical symptoms	Symptoms reported are diverse, from typical urinary symptoms to atypical. Perceived severity varies from severe and disabling to an “everyday nuisance” [38].		
Self-diagnosis of UTI recurrence	Most become “experts” at recognising UTI symptoms, but some find it challenging [38]. Women “hope” it is not a UTI, but often symptoms persist and worsen [34].		
Causes and/or triggers of rUTIs	Several potential causes and triggers for rUTIs are described. Women blame themselves for rUTIs and treatment failure and search for personal factors causing recurrence.	Various factors, including lifestyle, hormonal, environmental, and stress are described as potential causes or triggers for rUTIs. There is also fear of an underlying cause and consequence of recurrent UTIs, and women blame themselves for treatment failure and recurrence.	6
Anxiety and fear of rUTIs	Fear that “there must be something wrong” [23]. Fear of the consequences of rUTIs, from general concern to specific concerns, such a kidney damage or persistent urinary symptoms.		
Physical impact	The physical impact of rUTIs is generally described as severe. Attending healthcare multiple times also results in fatigue.	Recurrent UTIs generally have a significant and widespread effect on women’s quality-of-life. They impact women physically and psychologically and have significant impacts on their relationships, work, home, and social lives. However, there is variability between women on the impact of rUTIs, from none to severe.	8
Psychological impact	There are several significant negative psychological emotions at the time of UTIs. There are feelings of being helpless, frustrated, and anxious about future UTIs. Seeking healthcare results in stress and anxiety.		
Impact on relationships	Social contact is often reduced during acute UTIs, and women feel like a “burden” [34]. Recurrent have a significant impact on sexual relationships at the time of UTIs and can damage long-term relationships, due to intercourse being a known trigger.		
Impact on work/education/home life	Recurrent UTIs impact the ability to work at home and attend paid employment due to the UTI itself but also seeking healthcare. This has financial implications.		
Impact on social and leisure activities	Recurrent UTIs negatively impact interests and hobbies, and women are less likely to leave home. When they leave the house, extra precautions and planning are required.		
Variable impact on quality-of-life	There is variability of the impact of rUTIs, from severe to no reported impact on social or sexual activities.		
Use of Complementary and Alternative Medicine (CAM)	Those with an rUTI appear to commonly use CAM with variable success. CAM are used differently; some use them instead of antibiotics, whilst others use them in addition to antibiotics, and others use to prevent recurrence. Alternative treatments can be expensive.	Women with rUTIs commonly use self-help, lifestyle, and CAM options to try to prevent rUTIs with variable success. Self-care can help women feel empowered about managing their rUTI.	9
Lifestyle-management strategies	Increasing hydration is a common method used to reduce UTI recurrence, as well as increased personal hygiene and passing urine after intercourse. During active infections, a variety of measures are used. This appears to depend on their beliefs of the underlying cause.		
Over-the-counter medication/therapies	Those with rUTI often have tried a variety of over-the-counter medications, such as D-mannose, probiotics, cranberry drinks, and vitamin C.		
Empowerment	Self-help or use of CAM can offer self-control or empowerment.		
The effectiveness of antibiotics	Most with rUTIs find acute antibiotics effective, and long-term antibiotics can be “transformative”. Effects of acute antibiotics are short-lived and sometimes have minimal/no effect. Attitudes toward antibiotics vary.	The effectiveness of antibiotics varies from being effective short term to having minimal or no impact. Attitudes about their use vary, and there is fear and concern about their use.	7
Concerns and fears about antibiotics	There is fear and concern about antibiotic side-effects, the development of resistance, and impacts on their immune system increasing subsequent infections. Some describe they would resist taking antibiotics. Personal experiences of antibiotic resistance and treatment failure can result in self-blame and frustration.		
Consulting behaviour	Those with rUTI do not seek healthcare for all UTIs but do for the majority. Triggers include symptoms not settling, worsening pain, or a positive dipstick. There is a reluctance to attend healthcare services and feeling “guilty” for doing so. Some feel their experience of rUTIs allows more involvement in treatment discussions. Goal of treatment was not always full recovery but reduction in UTI frequency. The cost of care is barrier to care in the U.S.A. Some question the need to consult and undergo diagnostic tests since the diagnosis is clear.	Women with rUTIs often seek healthcare for UTI recurrence but describe anger and frustration about their care, the use of antibiotics, and underestimation of the impact of the rUTI.	7
Positive experiences of healthcare	Positive experiences of care include easy access and fast treatment with the “correct” medicine [34]. Positive experiences with HCPs were when they were knowledgeable about rUTIs and were supportive, listened to them, and willing to refer to secondary care. Relief at referral to secondary care and undergoing further investigation.		
Negative experiences and frustrations with healthcare	Negative experiences of care involved not receiving the “correct” treatment, a lack of information about treatment or prevention, waiting for urine results, and perceived lack of investigation [34]. Negative experiences with HCPs were being treated “nonchalantly”, with HCPs being unsupportive, dismissive, or unsympathetic; being patronising, not listening; and refusing referral to secondary care [34]. There is a lack of trust in HCPs if they do not listen with regard to previous treatment or ongoing management.		
Use of antibiotics by HCPs	There is a feeling that antibiotics are overprescribed and concerns of their use in the absence of infection. There is anger and frustration at HCPs not discussing antibiotic alternatives. Frustration of delayed antibiotic prescribing whilst awaiting culture results.		
Underestimation of the impact of rUTIs by HCPs	HCPs underestimate the effect of rUTI, leading to mistrust. This is exacerbated by gender differences.		
Lack of or need for more information on rUTIs	Those with an rUTIs want more information about the causes, treatment options, and preventative options.	There is a need for more information on the causes and treatment options, and this is sought via a variety of sources.	7
Seeking more information and sources of information	Advice and information are sought via a variety of means such as online communities, the internet, magazines, TV, and CAM practitioners but not relatives. Information from HCPs about self-help strategies include cranberry, fluid intake, and passing urine after sex but not the negative impacts of antibiotics.		
Support networks for women with rUTIs	Online chatrooms and web forums form a strong support network in terms of sharing experiences, recommendations, and knowledge.		
Need for more research	Those with an rUTIs feel that research is needed on non-antibiotic treatments and preventatives, and diagnostics. Patients want more accurate antibiotic prescribing.		

**Table 5 antibiotics-12-00434-t005:** Conceptual categories, 1st-, 2nd-, and 3rd-order constructs for healthcare professionals.

Concepts	1st- and 2nd-Order Construct Interpretations	3rd-Order Constructs	Number of Studies
The use of urine sampling	Most would send a urine sample for culture. Reasons for sending urine for culture are mostly for diagnosis and for referral, but also identifying unusual organisms or antibiotic resistance. Differences in opinion of interpreting urine dipstick result if it does not match patient symptoms.	There are differences in the use of and the reason for conducting urine sampling from making a diagnosis to identifying unusual organisms and resistance. Healthcare professionals (HCPs) also highlight the limitations of urine sampling, including a mismatch with symptoms and potential for “over-medicalisation” [22].	3
The benefits and limitations of urine culture	Benefits of urine sampling are to confirm infection, identify unusual organisms or resistance and ensure the correct antibiotic. Limitations of urine sampling include the cost, concerns of over-medicalising UTIs, and that it does not change initial management.		
Making the correct diagnosis and not missing serious disease	Making the correct diagnosis is important and not missing serious pathology. HCPs generally see patients with rUTIs in person to assess for pyelonephritis.	HCPs consider that there are numerous causes for rUTIs, including lifestyle and medical factors. Making the correct diagnosis of rUTIs is important, and HCPs worry about missing serious disease.	2
Causes and triggers for rUTIs	Potential causes of rUTIs include medical and lifestyle factors.		
The impact of rUTIs on patients	Awareness of rUTIs having a significant and wide impact on quality-of-life.	HCPs appreciate that rUTIs have a significant impact on quality-of-life and affect numerous areas of patients’ lives. The impact on patients also influences the provision of antibiotics.	1
Influence of the impact on management	Due to the impact of rUTIs, HCPs were more willing to prescribe immediate antibiotics. Provision of delayed antibiotics empower patients to control the impact of symptoms.		
Self-care advice given	Lifestyle changes are important in rUTI management. Confusion about the evidence for cranberry and cystitis sachets for prevention.	HCPs believe that self-help measures are important to prevent UTI recurrence, including hydration and hygiene advice. HCPs believe that more information is needed to improve self-management.	2
The use of self-care advice in rUTI	Self-help measures can be used to share responsibility with patients in the context of treatment failure.		
Lifestyle and behavioural changes and over-the-counter and natural remedies	Self-help preventative strategies include hydration, sodium bicarbonate, barley water, and cystitis sachets.		
Preventative information for patients	UTIs are overtreated, and more information on self-help management could help improve self-management.		
Views on complementary and alternative medicine (CAM)	There is interest in CAM, particularly in cases where conventional treatment had failed. CAMs are “biologically plausible” [22] but could be effective via placebo effect.	HCPs are interested in the use of CAM and feel that they could be effective. However, they have significant concerns about their safety and regulation, and their lack of knowledge prohibits them from recommending them.	1
Concerns about CAM	Concerns about the effectiveness, safety, regulation of herbal practitioners, quality assurance, and interactions with medication. Concerns about costs to patients and legal risk of recommending CAM.		
HCPs’ knowledge of CAM	Lack of knowledge about CAMs limits HCPs recommending them.		
When HCPs would consider CAM use	Use of CAM would be considered if further research confirms efficacy and safety and quality-control procedures.		
Effectiveness of antibiotics	Antibiotics are effective in the short term.	HCPs feel antibiotics are effective in the short term but have significant concerns about their use, including side-effects, developing resistance, and over-medicalisation.	2
Concerns about antibiotic use	Concerns about the use of antibiotics due to the risk of resistance, side-effects, over-medicalising a “mild, self-limiting condition” [22], hiding a serious diagnosis, or when no infection is present.		
Discussions with patients about the benefits and risks of antibiotics	HCPs would discuss the risk of bacterial resistance in patients with an rUTI.		
Further investigation and referral to secondary care	Attitudes toward referral for further investigation are mixed and vary for when they consider referral is required.	Attitudes towards when refer to secondary care for further investigation and its benefit varies between HCPs. Some refer based on specific criteria, whereas others base it on gut feeling. Some HCPs feel that further investigation has little impact on management.	2
Over-medicalisation	The provision of antibiotics for a potentially “self-limiting condition” [22] creates dependency and over-medicalisation.	HCPs must consider and balance several competing interests in rUTI treatment. These include a desire to improve patient’s symptoms and minimise antibiotic use due to concerns about resistance and limited, conflicting, or lacking guidance. patient expectation and concerns about “over-medicalising” [22] UTIs.	3
Lack of or conflicting advice	Need for more national guidance for the management of rUTIs. Conflicting information about cranberry and cystitis sachets in guidelines.		
Desire to improve patient’s symptoms	Desire to reduce patients’ symptoms, and HCPs express a range of negative emotions if this cannot be achieved.		
Risk of resistance	Guidance advises HCPs to reduce antibiotic prescribing to prevent resistance. rUTIs are a risk factor for resistance.		
Patient expectation	Patients’ expectations are changing, and they want to know what is causing their rUTI. HCPs perceive patients want and expect further investigation. Belief that patients expect antibiotics for urinary symptoms.		

## Data Availability

The data presented in this study are available on reasonable request from the corresponding author.

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
