# Peer review of "Patients’ and Healthcare Professionals’ Experiences and Views of Recurrent Urinary Tract Infections in Women: Qualitative Evidence Synthesis and Meta-Ethnography"

_antibiotics, 2023, doi:10.3390/antibiotics12030434_

Round 1
Reviewer 1 Report
v Strengths
The study has a sound methodology, clearly explained, aimed towards a very relevant subject. It’s mainly focused on the specific area of UK and there’s several reference cited towards that specific demographic. The methodology used to pursue this study is very well described, with detailed supplementary materials which provide useful details to understand this research. The topic addressed is relevant and clearly more research on the topic of Quality of Life of women with rUTI and HCP’s insight on it is needed.
v Weaknesses
Unfortunately conclusions on HCP’s perspectives on the matter are not as strong as patients’ one, as stated by the authors themselves. While the authors are well aware of the risks of selection biases or biases of other nature, this possibility can’t be overlooked in the part of the study regarding HCP’s. Still, the concepts emerged from this part of the study are generic and don’t claim to be very specific, so there’s an evident attention from the authors to the strength and weakness some of the data might have.
This study doesn’t appear to address specifically the different demographics and, thus, different risk factors and lifestyles among young adults and adults before and after menopause. This study provides a first starting point for further research, but perhaps a more strict age-related focus might help improve what issues women in different ages might feel and how they deal with it, how medical culture shiftes between women in their 20s and women in their 50s and this might improve confidence in concepts regarding CAM and self-diagnosis.
It would have been interesting to see if HCP’s have made some claims about why might be hard for them to manage rUTIs, especially considering finding n° 15 – “HCPs must consider and balance several competing interests when managing rUTI”. In the studies analysed by the authors, were there some hypothesis made by the interviewed professionals? Like, access difficulties to higher levels of care, or unstandardized practice among different professionals etc. It would be a very interesting topic to tackle.
TITLE
It’s very explicative. I’d suggest to add “in women” after “urinary tract infections”, considering this study is addressing women specifically.
GLOSSARY
A glossary might be helpful to improve readability.
INTRODUCTION:
Have you considered tackling briefly how women with different ages are exposed to different risk factors? Do you think it might improve the understanding of readers about the topic?
TABLE 1
In exclusion criteria, have you considered also immunocompromised patients?
TABLE 3
Please consider adding some space between column “Explanation of GRADE-CERQual Assessment” and “References” to improve readability, considering right now the content of these two columns are very close to each other and it might be hard to read at first glance.
SUPPLEMENTARY TABLE 1: step 14, page 11-29, page 11 is the last part of table 3. Please correct this reference by adding the whole table 3 or starting from page 12.
v Suggestions to Author/s
Discussion: The perceptions of HCPs towards rUTI in women may vary based on their competences. In Napolitani et al. 2022 “How to Improve the Drafting of Health Profiles.” Int. J. Environ. Res. Public Health 2022, 19, 3452. https://doi.org/10.3390/ijerph19063452, as part of the study about the profiling of different patients with a standardized index such as CIRS might have different perception of single cases based on their competences and experiences, but genitourinary diseases presented a significant agreement between the two specialties, further improved after training in patients profiling.
The usage of standardized health profiling indexes (such as CIRS, but there are many others) might help to see if opinions of Health Practitioners are homogeneous. If you’re considering further developing this line of research, having a comparison with these validated index might shed a light on how different doctors perceive women with rUTI’s quality of life.
Reviewer 2 Report
Review of recurrent UTI, qualitative synthesis.
General comments:
1. This is a comprehensive large study on an important clinical topic that has not been well covered in the literature and guidelines previously
2. The authors have used a thorough methodology, and have well addressed most of the topics related to the performance of the review. They have also used well established instruments to qualify their data extraction.
3. The authors have found many important findings in their extensive reporting in both GPs and patients. Their findings may have important clinical implications.
Specific points for discussion:
1. Definitions of rUTI, this could be discussed more with the international definition of 2 last half year or 3 last year as reference. In our own study on methenamine, we chose to have 2 years of follow up, to avoid misclassification of rUTI. (1) This seems not to be an important bias in this review, but should be mentioned.
2. Among the GPs I miss more discussion on AMR and the possible consequences In relation to rUTI
3. As to the treatment, methenamine should be mentioned as it seems to have no resistance driving effect and few side effects. It has been heavily used in the Nordic countries. There are two large ongoing studies, one from primary care and one from secondary care (2,3).
1. Preventive effect of methenamine in women with recurrent urinary tract infections - a case-control study. Rui L, Lindbaek M, Gjelstad S.Scand J Prim Health Care. 2022 Sep;40(3):331-338. doi: 10.1080/02813432.2022.2139363. Epub 2022 No
2. Methenamine hippurate compared with antibiotic prophylaxis to prevent recurrent urinary tract infections in women: the ALTAR non-inferiority RCT.Harding C, Chadwick T, Homer T, Lecouturier J, Mossop H, Carnell S, King W, Abouhajar A, Vale L, Watson G, Forbes R, Currer S, Pickard R, Eardley I, Pearce I, Thiruchelvam N, Guerrero K, Walton K, Hussain Z, Lazarowicz H, Ali A.Health Technol Assess. 2022 May;26(23):1-172
3. Methenamine hippurate to prevent recurrent urinary tract infections in older women: protocol for a randomised, placebo-controlled trial (ImpresU). Heltveit-Olsen SR, Sundvall PD, Gunnarsson R, Snaebjörnsson Arnljots E, Kowalczyk A, Godycki-Cwirko M, Platteel TN, Koning HAM, Groen WG, Åhrén C, Grude N, Verheij TJM, Hertogh CMPM, Lindbaek M, Hoye S. BMJ Open. 2022 Nov 1;12(11):e065217. doi: 10.1136/bmjopen-2022-065217
